



# Monitoring the Earth's deformation with the SPOTGINS series

Alvaro Santamaría-Gómez[1,2], Jean-Paul Boy[3], Florent Feriol[1], Médéric Gravelle[4], Sylvain Loyer[5], Samuel Nahmani[6,7], Joëlle Nicolas[8], José Luis García Pallero[9], Aurélie Panetier[6,7], Arnaud Pollet[6,7], Pierre Sakic[6]

[1] Geosciences Environnement Toulouse, Université Paul Sabatier, CNES, CNRS, IRD, UPS, Toulouse, France.
[2] Centre National d'Etudes Spatiales, Toulouse, France.
[3] Institut Terre-Environnement Strasbourg, Université de Strasbourg, ENGEES, Strasbourg, France.
[4] Littoral Environnement et Sociétés (LIENSs), CNRS, La Rochelle University, La Rochelle, France.
[5] Localisation & Orbitographie, Collecte Localisation Satellite, Ramonville Saint-Agne, France.
[6] Université Paris Cité, Institut de physique du globe de Paris, CNRS, IGN, Paris, France.
[7] Université Gustave Eiffel, ENSG, IGN, Paris, France.
[8] Laboratoire Géomatique et Foncier, Cnam, Le Mans, France.
[9] ETSI en Topografía, Geodesia y Cartografía, Universidad Politécnica de Madrid, Madrid, Spain.

*Correspondence to*: Alvaro Santamaría-Gómez ([alvaro.santamaria@cnes.fr](mailto:alvaro.santamaria@cnes.fr))

**Abstract.** A distributed Global Navigation Satellite System analysis center, designated SPOTGINS, has been established by several research groups that utilize the GINS software and the CNES-CLS precise products. Despite the heterogeneity in their research objectives, the SPOTGINS members apply the same configuration and metadata. The computed global ambiguity-fixed precise point positioning time series are fully consistent among the members, and are subsequently published as a single product. This product facilitates a range of research activities, including but not limited to the precise monitoring of the Earth's

deformation and the water vapor content of the troposphere. A comparison of the SPOTGINS series with published series from the Nevada Geodetic Laboratory solution shows no significant difference in quality.



# 1 Introduction

The *Shared and Operational PPP Solutions Processed with GINS* (SPOTGINS) is a novel initiative based on a distributed
Global Navigation Satellite System (GNSS) analysis center where independent research groups cooperate by using the same
software, the same processing strategy, and the same metadata to generate a common set of GNSS time series. The primary
objective of this initiative is the generation of daily global Precise Point Positioning (PPP) position time series, as well as
hourly zenith tropospheric delay (ZTD) time series. These series allow monitoring the Earth's deformation and the atmospheric
water vapor content at the millimeter level through the 21st century. The SPOTGINS position time series are currently available
on The Geodesy Plotter of the Solid Earth Center portal (ForM@Ter)[1]. The ZTD series will be available on the same portal in
the near future.

The SPOTGINS cooperative was established in 2022 following the third reprocessing campaign of the International GNSS
Service (IGS; Johnston et al., (2017)), when several research groups in France started to produce ambiguity-fixed GPS and
Galileo PPP position time series with the GINS software (Michel et al., 2021; Nicolas et al., 2021), and the precise orbit, clock
and phase biases computed by the Centre National d'Etudes Spatiales (CNES) - Collecte Localisation Satellites (CLS) IGS
analysis center (Loyer et al., 2012). The research groups decided to unite into a collaborative processing effort with the support
of the CNES-CLS analysis center. Each SPOTGINS member pursues distinct research objectives related to the Earth's
deformation or the tropospheric water vapor, yet all contribute to the common processing by providing the series of a chosen
set of GNSS stations depending on their geographic location, network label, or research project. By applying the same
processing strategy, the obtained series are fully consistent and interchangeable among the members. At the time of writing
(April 2025), the current members are described in Table 1. Figure 1 illustrates the distribution of the SPOTGINS sub-networks
processed by each member. The total number of stations is 5768 as of April 2025.

The processing strategy is based on the zero-differenced ionosphere-free ambiguity-fixed carrier phase and code observations
from the GPS and Galileo constellations. The models and corrections applied are described below and are fully consistent with
the strategy used by the CNES-CLS IGS analysis center to compute the precise orbit and clock products. This avoids any
relative range bias with respect to the fixed orbit and clock products, which increases the quality of the computed PPP series.
Consequently, SPOTGINS can be regarded as the PPP densification of the CNES-CLS network solution aligned to the IGS20
reference frame (Rebischung et al., 2024).

In addition to a common processing strategy, the SPOTGINS members also share the station metadata. This metadata includes
the following: station reference coordinates, receiver and antenna models, antenna eccentricity, antenna orientation, ocean tide
loading coefficients, and co-seismic station displacement predictions from Métivier et al., (2014). Each member is responsible
for providing the full history of each station's metadata, and also for keeping it up to date, within their respective sub-networks.

---

[1] https://www.poleterresolide.fr/geodesy-plotter-en/#/?solution=SPOTGINS (accessed April 2025)



The consistency of the series computed by each member is periodically validated by the intercomparison of a small set of stations processed by all members.

**Table 1. List of members participating in SPOTGINS in April 2025.**

| Acronym | Member name |
|---------|-------------|
| EOST | Ecole et Observatoire des Sciences de La Terre, Institut Terre et Environnement de Strasbourg<br>https://eost.unistra.fr, https://ites.unistra.fr |
| ESGT | Ecole Supérieure des Géomètres et Topographes, Laboratoire Géomatique et Foncier (GeF)<br>https://www.esgt.cnam.fr/recherche |
| IPGP | Institut de Physique du Globe de Paris<br>https://www.ipgp.fr/en |
| OMP | Observatoire Midi-Pyrénées, Géosciences Environnement Toulouse<br>https://www.get.omp.eu |
| ULR | Université de La Rochelle, Littoral Environnement Sociétés<br>https://lienss.univ-larochelle.fr |
| UPM | Universidad Politécnica de Madrid, Escuela Técnica Superior de Ingenieros en Topografía, Geodesia y Cartografía<br>https://www.topografia.upm.es |

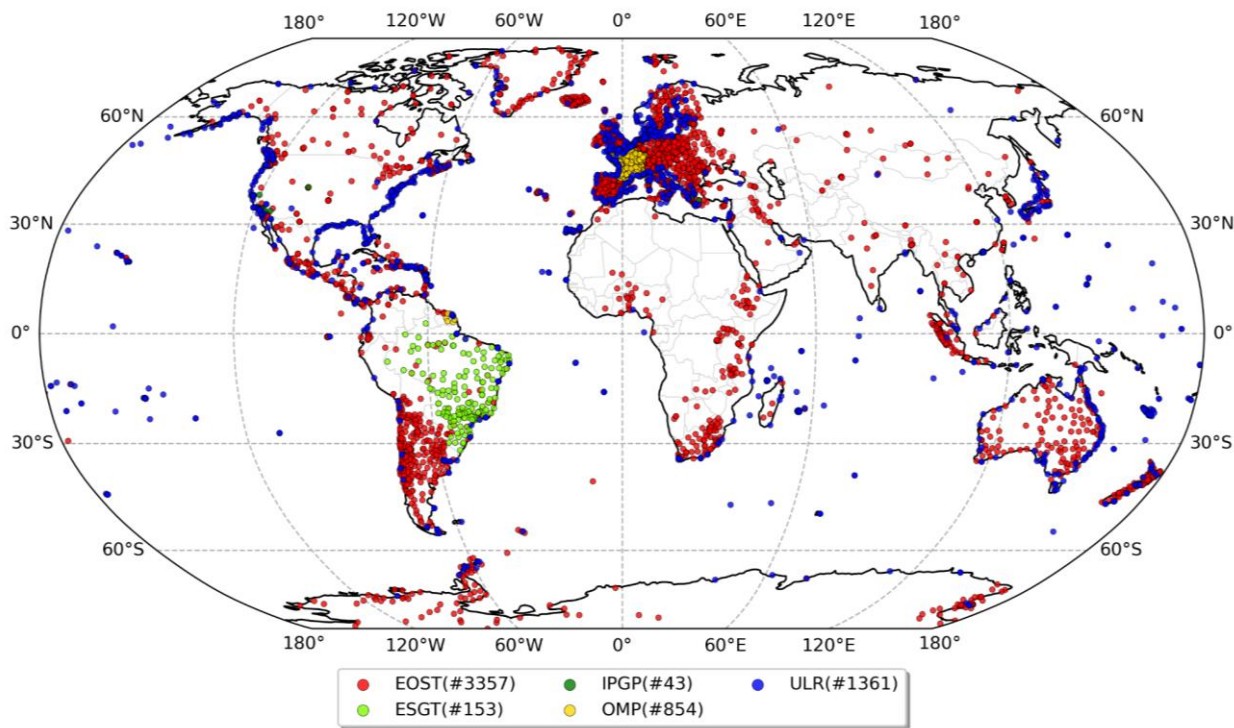

**Figure 1: The SPOTGINS sub-networks processed by each member in April 2025. The number of stations included in each sub-network is also indicated.**

## 2 GNSS data processing

The PPP processing is based on ambiguity-fixed carrier phase and code observations from the GPS constellation, since may 2000, and the Galileo constellation, since October 2018. These dates correspond to the availability of the ambiguity-fixed CNES-CLS precise products for each constellation. The observations are sampled at 5 min, and a cut-off elevation angle of 8 degrees is applied to minimize errors caused by multipath, atmospheric propagation, and receiver/satellite antenna phase patterns.

The input observations are screened for quality and their uncertainty (s) is assigned by an empirical elevation-dependent (e) function based on stacked long-term post-fit residuals. This function takes the following form:

$$s = \frac{s_0}{a+(1-a)\,\sin(e)} \tag{1}$$

where $s_0$ is the observation uncertainty at the zenith with values of 3.5 and 600 mm for phase and code observations, respectively, and $a$ is the amplification term with a value of 0.15. The elevation-dependent uncertainties obtained are then scaled by the relative precision of the computed orbit for each individual satellite.



Phase observations are corrected for wide-lane satellite-dependent biases computed weekly by the CNES-CLS analysis center[2], which, together with the associated daily integer satellite phase clock biases, allow PPP users to perform ionosphere-free integer ambiguity resolution for each GNSS station (Laurichesse et al., 2009). Similarly, to comply with the GPS P1/P2

convention of the IGS products, code observations are corrected, when necessary, for satellite-dependent monthly differential code biases computed by the Centre for Orbit Determination in Europe.

Satellite-dependent antenna phase center offsets (PCO) in the nadir direction and block-dependent horizontal PCO corrections are applied using the IGS20 antenna calibration model. Satellite block-dependent nadir angle-dependent absolute phase center variations (PCV) are corrected using the same IGS20 model. For the receiver antenna, absolute PCO and direction-dependent

PCV corrections are also applied using the IGS20 model. All PCO and PCV corrections are frequency-dependent. The receiver antenna PCV corrections are rotated according to the antenna orientation indicated in each station's sitelog file.

Phase observations are corrected for the wind-up effect (Wu et al., 1993) by taking into account the satellite attitude using the nominal yaw model for GPS (Bar-Sever, 1996), and the nominal attitude law for Galileo released by the EU Agency for the Space Programme (EUSPA)[3]. The CNES-CLS satellite clocks are corrected for second-order relativistic effects due to the

small orbit ellipticity of the GPS satellites. Furthermore, a daily receiver clock phase bias is removed between the GPS and Galileo observations.

Signal path delays due to the propagation through the neutral atmosphere are corrected using the VMF1 mapping function grids (Boehm et al., 2006), which include zenith hydrostatic and wet delays from the European Centre for Medium-Range Weather Forecasts (ECMWF) reanalysis. The zenith wet delays are adjusted using a piecewise linear function at 1-hour

intervals, together with two horizontal gradients per day (Chen and Herring, 1997).

Signal path delays due to the propagation through the ionosphere are accounted for, at first-order, by forming the ionosphere-free linear combination of the L1/L2 GPS and the E1/E5a Galileo frequencies. Second-order ionospheric delays are corrected using the vertical total electron content values extracted from the daily IGS Final Global Ionosphere Maps (Hernández-Pajares et al., 2011).

Station displacements due to the solid Earth, solid Earth pole, and ocean pole tides are corrected using the 2010 International Earth Rotation and Reference Systems Service conventions (Petit and Luzum, 2010), including the latest linear mean pole model. The solid Earth tide correction also includes the permanent term, which corresponds to a conventional tide-free frame. Station displacements due to ocean tide loading are corrected using predictions for the 11 main tidal constituents extracted from the FES2014b (Lyard et al., 2021) model, which are then completed by interpolating the tidal admittances. Displacements

due to the atmospheric thermal tides loading and the non-tidal loadings are not corrected at the observation level.

---

[2] https://igsac-cnes.cls.fr/html/products.html (accessed April 2025)

[3] https://www.gsc-europa.eu/support-to-developers/galileo-satellite-metadata (accessed April 2025)

A summary of the SPOTGINS processing strategy is available at the Solid Earth Center portal (ForM@Ter)[4]. This file may change in the future to reflect changes in the processing strategy with respect to the description given above.

## 3 Comparison of the SPOTGINS series

The SPOTGINS position time series have been compared to published series from the Nevada Geodetic Laboratory (NGL; Blewitt et al., 2018). The NGL is a global PPP solution obtained with the Gipsy software and the JPL IGS precise products. Figure 2 shows an example of the difference of position time series obtained from these two solutions for the same station. The ITRF2020 plate motion model (Altamimi et al., 2023) was removed from both solutions for visualization purposes. The vertical jump near 2019 is here likely caused by the use of wrong station metadata in the NGL solution.

The dispersion of the detrended and cleaned series by solution and coordinate component is shown in Fig. 3, together with the number of common stations. The same period was considered for each of the 2948 pairs of common stations considered. The typical dispersion of both the SPOTGINS and NGL solutions is at the level of 2 mm and 6 mm, for the horizontal and vertical components, respectively.

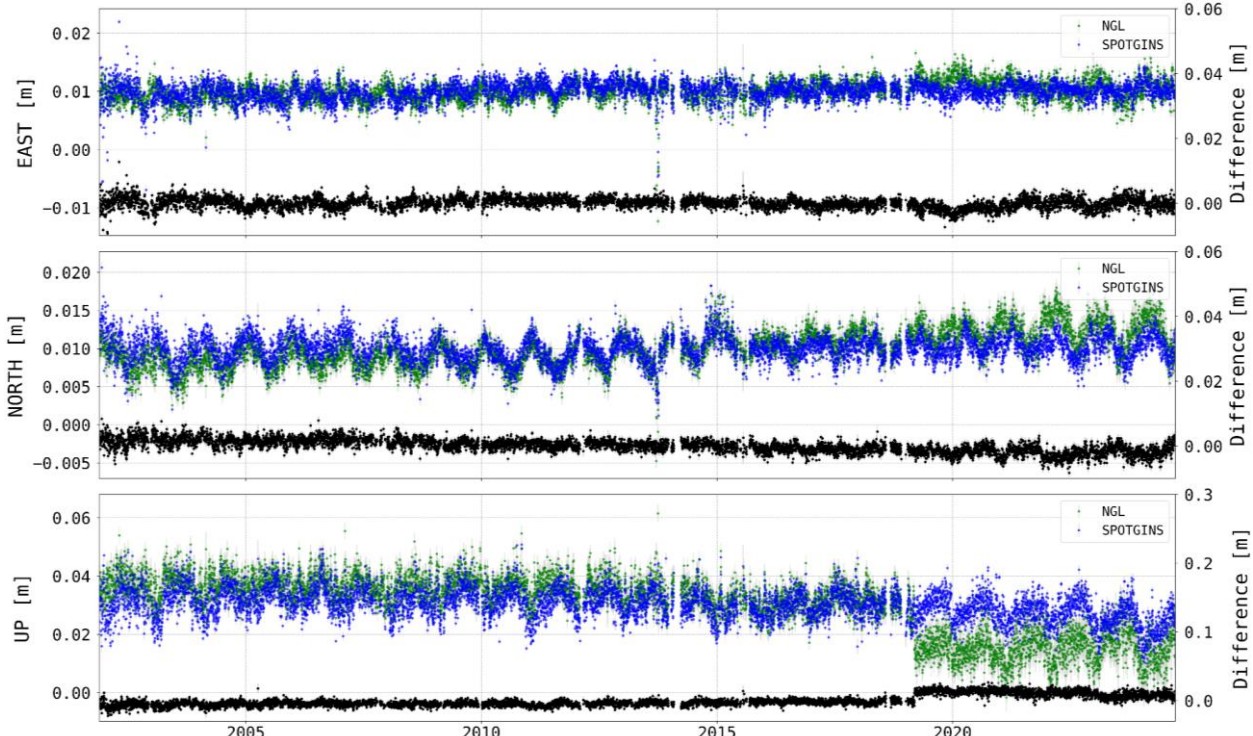

**Figure 2: Time series of daily displacements from the SPOTGINS (in blue) and NGL (in green) solutions for the LROC00FRA station with respect to the Eurasian plate. The daily differences between both solutions are represented in black (right y-axis).**

---

[4] https://www.poleterresolide.fr/geodesy-plotter-en/#/solution/SPOTGINS (accessed April 2025)




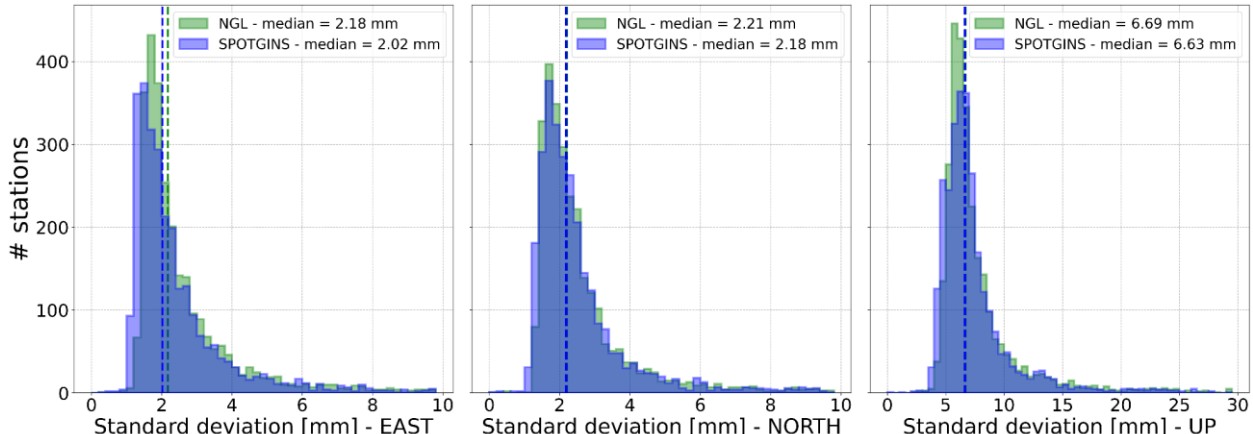

**Figure 3: Time series of daily displacements from the SPOTGINS (in blue) and NGL (in green) solutions for the LROC00FRA station with respect to the Eurasian plate. The daily differences between both solutions are represented in black (right y-axis).**

To assess the quality of the terrestrial frame realized by the SPOTGINS position series, we compared the terrestrial frame
defined by the estimated positions of 410 IGS stations included in the SPOTGINS solution, day by day, to the ITRF2020 reference frame. We estimated the daily translations, rotations and scaling factors between the SPOTGINS solution and the ITRF2020. As the CNES-CLS orbit and clock products used to compute the SPOTGINS solution are referenced to the IGS20 frame, the obtained SPOTGINS series must also be referenced to the ITRF2020 frame, i.e. no net translation, rotation and scale change should exist between both frames. Small departures from the ITRF2020 are expected due to the different number
of stations used in alignment of the CNES-CLS products, which varies with time.

Figure 4 shows the estimated time series of the daily transformation parameters and the number of stations used in the alignment. The mean bias and drift of each transformation parameter are shown in Table 2. All transformation biases and drifts are smaller than 1 mm and 0.1 mm/year, respectively, confirming the excellent quality of the referencing of the SPOTGINS solution.



**Figure 4: Daily time series of the transformation parameters between the SPOTGINS solution and the ITRF2020, and the number of common stations used for the computation. Translation and scale factor in millimeters, rotation in micro-arc seconds.**



**Table 2. Bias and drift of the transformation parameters between the SPOTGINS solution and the ITRF2020.**

|  | **TX** (mm) | **TY** (mm) | **TZ** (mm) | **SC** (mm) | **RX** (mm) | **RY** (mm) | **RZ** (mm) |
|---|---|---|---|---|---|---|---|
| **Bias** | -0.54 +/- 0.01 | 0.08 +/- 0.01 | -0.01 +/- 0.01 | 0.02 +/- 0.01 | 0.09 +/- 0.01 | 0.38 +/- 0.01 | -0.31 +/- 0.01 |
| **Drift** | 0.01 +/- 0.00 | 0.00 +/- 0.00 | 0.01 +/- 0.00 | 0.08 +/- 0.00 | 0.00 +/- 0.00 | 0.02 +/- 0.00 | -0.02 +/- 0.00 |


## Data availability

The SPOTGINS position time series are available from ForM@Ter, the French National Solid Earth Center portal (https://www.poleterresolide.fr/geodesy-plotter-en/#/?solution=SPOTGINS, accessed April 2025) under CC-BY license at https://doi.org/10.24400/170160/20250414 (Santamaría-Gómez et al., 2025).

The GNSS data used to compute the SPOTGINS series are available from the data servers included in the supplemental material. The ORPHEON GNSS RINEX data are provided for scientific use in the framework of the GEODATA-INSU-CNRS convention.

## Author contributions

ASG, JPB, FF, MG, SL, SN, JN, APa, APo and PS processed the GNSS data. SL provided technical support with the GINS
software and the CNES-CLS precise products. MG provided the results of the comparison analysis. APo provided the results of the alignment analysis. ASG wrote the manuscript with corrections provided by FF, MG, SL, JN, JLPG, APo and PS. MG created the figures.

## Competing interests

The authors declare that they have no conflict of interest.

## Acknowledgements

The authors are thankful to the CNES and CLS space geodesy teams for their support with the GINS software, and to Elisabeth Pointal and the Geodesy Plotter team for their support with the publication of the series. All the past, present and future members of SPOTGINS are acknowledged for their past, present and future contributions to this cooperative effort. Especially, Alexandre Michel is acknowledged for developing the initial file structure, the name and logo of SPOTGINS.


**Financial support**

SPOTGINS is financially supported by CNES as an application of the IGS-AC, DEFRHEO, and GEOSPARC projects. IPGP's processing is performed on the S-CAPAD/DANTE numerical computations platform.

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
