# Peer review of "Monitoring the Earth's deformation with the SPOTGINS series"

_Earth System Science Data, 2025_

## Author Response (AR1)

**Answer to RC1:**

We are very thankful to Matt King for his constructive review and comments, which will improve the clarity of the final version of this manuscript. Each raised comment (highlighted in blue) is addressed below.

The authors present a new global GNSS dataset that provides time series of Earth deformation and tropospheric water vapour at locations globally. It is a novel (for GNSS) analysis approach in that it is a cooperative and dsitributed model of data analysis using consistent processing schema. The data products will be useful for many communities interested in Earth deformation studies and water vapour.

The methods employed are state of the art and generally fully described. Assessments of the coordinate time series against those from other solutions are provided, demonstrating a high level of consistency. No comparison is provided for the water vapour products. I could not see how to find the water vapour products on the website. So I suggest water vapour should be removed from the manuscript unless a fuller description is provided.

The water vapour series are still under validation and not published yet. These products are not discussed in the manuscript. We will remove any reference to the water vapour products in the final version of the manuscript.

The data are provided on a web platform but not on a permanent data repository. It does not seem a requirement of ESSD to have the data somewhere permanent. That is ok, but it does raise the issue of versioning, and access to archived versions (I return to that below).

The current GNSS dataset has an operational status, with new stations continuously being added and the series being increased over time. The current dataset is to be archived in a permanent data repository and made accessible to the users only when a new reprocessing of the whole dataset is necessary. At that point, the new operational dataset will be made available on the same web platform. This information will be included in the final version of the manuscript.

When I downloaded the data in .enu file format the data provide a header file but there does not seem to be any versioning of the datasets. This makes the data prone to erroneous analysis and comparison, and I strongly urge the authors to include versions for the their datasets and a changelog of kinds.

The versioning of the dataset is specified in the header of the .enu series files. For this, the header includes three elements: 1) the type of series. At this time, only position series are being published. The tropospheric series will be made available in the future. 2) the version of the series file (version 2 at this time), which allows for new metadata to be included in the header and/or in the number of columns of the file. 3) the GNSS products used (G20/GRG at this time). As we describe in the manuscript, there is full consistency between the strategy followed to compute the GNSS products by the CNES-CLS IGS AC and the PPP series of this dataset. In case the strategy of the GNSS products changes in the future, the label of the series header will change accordingly. Also in that case, a new version of the dataset will be made available after the reprocessing of the PPP series, which then will become the operational dataset as described earlier. This information will be included in the manuscript at the end of Section 1 for clarification and proper handling by the user.

The Headers do not include units,s and this should be resolved. The header columns are somewhat obvious to an expert but they are not described (e.g., jjjj.jjjjjjj is meant to refer to decimal Modified Julian Days it seems).

We agree with the reviewer's comment and we will change the "jjjj" and "yyyy" column names by more descriptive names, such as "modified Julian day" and "decimal year", respectively. These

changes can be applied to the series header without changing the rest of the series content. A new version 3 of the series files will be created. Other than the epoch columns, the units of the series values are already indicated in the file header.

I found the comparison to the NGL solutions helpful, although I am concerned that the comparison to just one station time series is prone to cherry picking. Please include some statistics for a few hundred randomly chosen sites, globally distributed. The 410 IGS sites would be a sensible selection.

Figure 2 shows the series comparison from one single station. This particular station was chosen to highlight the differences the reviewer points out below. Figure 3 shows the statistics of the series comparison from 2948 stations. This is indicated in the text of the manuscript, but unfortunately, the legend of Fig. 3 was mistakenly replaced by the same legend of Fig. 2. The legend of Fig. 3 will be corrected in the final version of the manuscript.

Otherwise, I have only a series of more minor comments, but I think most are essential nonetheless.

Abstract: The abstract should mention the temporal sampling of the data and the start date, the present end date of the time series and the number of stations considered at present.

We agree with the reviewer's comment and we will included this information in the final version of the manuscript.

**L49 it is not clear if metadata is a part of the dataset described and made available. please clarify this.**

The metadata concerning the station reference coordinates, receiver and antenna models, antenna eccentricity, and antenna orientation is also made available together with the position series. This information will be included in the final version of the manuscript in Section 1 and in the "data availability" section.

L61: capitalise May.

Done.

L67: "based on stacked long-term post-fit residuals" is jargon, especially, 'stacked'. It would be useful to know how many stations were involved and how many days/years. or some other guidance. i was surprised that the values are fixed across all sites (but not satellites). it would be interesting to see s vs e plotted in supp material and if this is purely an empirical relationship or if there is a basis for it in physics.

The relationship between "s" and "e" is purely empirical. However, the relationship is controlled by the amplification "a" parameter that captures the average change in the measurement noise with elevation, which is a known physical phenomenon. This weighting law was established in the GINS software long ago while trying to minimize the perturbation of the tropospheric mismodeling errors in the measurement noise. Within the IGS ACs groups, it is common to use a fixed weighting law. Only a few use an adjustable law, but all of them are empirical and based on an a priori fixed law. The fixed weighting law we use was obtained by the CNES-CLS team and was evaluated to be an improvement with respect to the 1/sin(e) and 1/sin(e)^2 laws that are commonly used by other groups. Unfortunately, there is no reference available on this evaluation. These two sentences will be rephrased in the final version of the manuscript.

**L79 for the present release you should specify the antex file version.**

As indicated in the manuscript, we use the IGS20 antex model. As for the specific version of this model, we use the same version as the one used by the IGS CNES-CLS AC for computing the daily GNSS products. The IGS20 antex model is updated regularly to include new antenna calibrations. The IGS CNES-CLS AC always uses the newest version available for the operational products, and

therefore, it is not possible to indicate a specific version of the IGS20 antex model in our dataset. This information is available in the header of the CNES-CLS SP3 orbit files from the IGS data servers.

L85 are the orbits in ITRF2020 or is this (as often in GIPSY) a fiducial free orbits and clocks and then a transformation later?

As indicated in the manuscript, the IGS CNES-CLS products are aligned to the IGS20 reference frame. No further transformation is needed for the SPOTGINS series.

L89 it would be good to tighten up on what is exactly meant by ECMWF reanalysis - is this ERA5 or something else (and please add the reference)

The VMF1 grids are based on the ERA-40 reanalysis and on the operational products afterwards. This information will be added to the final version of the manuscript. The reference of the VMF1 grids with a full description of how they are computed is already indicated in the manuscript.

L98 please clarify the origin of these OTL corrections (CM or CF) as consistent with the orbits and clocks (probably reference Fu and Freymueller in J Geodesy).

The SPOTGINS PPP series are fully consistent with the CNES-CLS GNSS products, which are aligned to the IGS20 frame. Therefore, the OTL corrections applied are computed in a CF frame. This was not included in the manuscript, but we agree with the reviewer's comment and this will be added in the final version to avoid any confusion by the non-expert user.

L101 I hope the full metadata is in the released dataset. please add that here.

The data availability section will include a reference to obtain the metadata that is also made available with this dataset.

Section 2: I did not see any mention on the temporal resolution of the coordinates or clock terms. Please clarify if this is part of a filter of sorts and provide the details on process noise if relevant (plus for tropo and gradients). Also, does the GINS metadata allow for non-north-pointing antennas?

As indicated in the manuscript (sections 1 and 3), the position series are estimated daily. The receiver clock terms are reduced in the normal equation system, which is equivalent to as if they were estimated on an epoch-by-epoch basis (5 min), but with a much smaller system to handle. No filter/strong constraint is applied on any parameter. This information will be added to the final version of the manuscript. The orientation of the ground antennas is taken into account, this is also already indicated in the manuscript (sections 1 and 2).

L108 explain the likely wrong metadata more. I presume this is antenna model? NGL use the header so you should be able to be fairly certain. Also, why are the two solutions drifting relative to one another in E and U? please confirm the frame of the NGL solution which I think may be IGS14? The difference in clocks and orbits should also be mentioned.

Position offsets like this, affecting mostly the vertical component, are typically caused by wrong metadata and antenna changes. Unfortunately, we lack of enough information to explain the offset in the LROC series of the NGL solution. All we can say for certain is that there is no change in our metadata, nor in the header of the RINEX files we use. The apparent drifts in the E and U components may be explained by several factors, including the differences of the reference frame and the way each solution realizes that reference frame, but also differences in the processing software and in the GNSS products. Small drifts may exist, but also small position offsets that could be interpreted as a drift. This information will be included in the final version of the manuscript.

Figure 4. the low-frequency signal in the scale probably deserves more of a mention, although I guess this pertains to orbits and clocks rather than the dataset being described here.

All the frame parameters are fixed by the GNSS products. The scale variation over time is an issue

that will be investigated by the CNES-CLS team independently of the SPOTGINS dataset. The CNES-CLS GNSS products are expected to evolve and improve in the future, as will do the products of any other IGS AC. As indicated above, whenever there is a change in the strategy used to compute the CNES-CLS GNSS products, a new version of the SPOTGINS dataset will be released.

There are several footnotes in the document but I could not see them linked in the text.

All the footnotes in the manuscript were revised and will be corrected if necessary in the final version.

**Answer to RC2:**

We are very thankful to Anna Klos for her review and positive comments. Each raised comment (highlighted in blue) is addressed below.

The authors mention that in addition to the position time series, they will also make available the ZTD series. I would advise not to mention this until the ZTD time series are ready and available. See the answer to RC1 on a similar comment.

4. I suggest that the article includes more characteristics of the new dataset, such as their comparison with the NGL data, which is shown for one station. It would be good to make comparisons for more stations,

See the answer to RC1 on a similar comment.

not just in the sense of standard deviation. Explaining the similarities and differences, including spatial patterns, would greatly help users understand the quality of the dataset presented.

We take note of this comment and we agree this would be a relevant comparison. However, including that amount of detail would require a level 3 dataset for both the NGL and SPOTGINS series. It would certainly provide valuable information on the current SPOTGINS dataset. However, we are not releasing such a level 3 dataset at this time, as it would require its own release notes (strategy, metadata, versioning, etc.).